# Investigating the Effects of Recycled Plastic as Fibers on Bending Behavior of Green Concrete Beams Exposed to Marine Environment

**DOI:** 10.3390/ma16175912

**Published:** 2023-08-29

**Authors:** Mohsen Ali Shayanfar, Hamid Shahrabadi

**Affiliations:** School of Civil Engineering, Iran University of Science and Technology, Tehran 16765-163, Iran; hamid_shahrabadi@civileng.iust.ac.ir

**Keywords:** green concrete, recycled fiber, metakaolin, zeolite, maximum load capacity, flexural toughness

## Abstract

Due to the noticeable production of greenhouse gases in cement production processes around the world, the use of supplementary cementitious materials (SCMs) like metakaolin/zeolite and the production of green concrete is inevitable, which leads to reducing the amount of environmental pollution and, specifically for maritime environments, improving the mechanical qualities of concrete. In addition, nowadays, the increasing use of plastic materials such as disposable glasses is considered a major problem in environmental pollution. Thus, using metakaolin/zeolite as an SCM and disposable glasses as fibers in concrete production may reduce environmental pollution and improve concrete’s properties. To do so, in this paper, the flexural behavior of green concrete beams containing metakaolin/zeolite at 10 and 20% as SCMs at 28, 90, and 180 days in the Oman Sea tidal environment was examined by studying the effects of utilizing 0.5 and 1% disposable-glass fibers in ring and strip forms. The findings demonstrate that ring (RFs) and strip fibers (SFs) in green concrete reduce a beam’s maximum load capacity (P_max_) by 31%, while RF and SF enhance green concrete beam flexural toughness by 8–20 times. Furthermore, the SF green concrete beams had 24% greater flexural toughness than RF beams at all ages. Finally, by improving the microstructure (by adding SCMs) and flexural behavior of marine concrete structures, in addition to increasing the load capacity and ductility of marine structures, the cracking and penetration of ions decreases; thus, the service life of the structures will increase.

## 1. Introduction

Concrete is a composite material obtained by curing a mixture of water, aggregates, and cement. The main reasons it is second only to water in worldwide use are its desirable mechanical properties and durability, long service life, acceptable physical and chemical properties, and availability of raw materials [1,2,3].

With the rapid increase in world population and the need for new building structures with different applications, the amount of concrete production and consumption in construction projects has recently been increasing rapidly. Building the various types of marine concrete structures with proper durability is essential because of the large population of people living near seas and the widespread development of infrastructure and various industries that must be built in aggressive environments such as the sea [4].

Despite the advantages and necessity of using concrete, the environmental impacts of providing its primary sources and their use, concrete’s poor performance against tensile forces, and its brittleness are the main disadvantages of conventional concrete [3,5].

For example, the preparation of raw materials for concrete mixtures is associated with high energy consumption and the wasting of several of earth’s natural resources, thus generating various types of environmental pollution. For one thing, the production process of cement, as the only factory product used in the production of concrete, creates a great deal of pollution. To be more exact, for the production of each ton of cement, between 0.7 and 1 t of carbon dioxide (CO_2_) is generated. About 5–7% of the total CO_2_ emitted in the world, one of the most important greenhouse gases, comes from the cement production process [1,4,6].

Therefore, it seems that reducing cement consumption by replacing it with natural materials (pozzolans) or other recycled materials is a suitable method for reducing environmental pollution, energy consumption, and economic efficiency [4]. The most common alternatives for cement are natural minerals (natural pozzolans) such as metakaolin, zeolite, and pumice; recycled materials (wastes) such as rice husk ash (RHA); and industrial by-products (synthetic) like silica fume, fly ash, and ground granulated blast-furnace slag (GGBS) [1,4].

As a result of the finer sizes of SCMs in concrete mix design, the size of the voids in the concrete and resultantly its porosity and permeability are reduced, leading to an increase in the concrete’s density. In addition, the use of SCMs improves the performance of fresh concrete, its mechanical characteristics, and durability, hence reducing the shrinkage and creep of the concrete. Thus, utilizing metakaolin and zeolite instead of cement to make green concrete reduces environmental impacts and improves mechanical properties and durability [1,7,8,9].

The crystalline structure of zeolite, as a type of aluminosilicate mineral with a volcanic-sedimentary origin, is very regular, has very small pores and channels, and can absorb or excrete 30% of its dry weight due to its huge specific surface area [1,10].

Metakaolin, a highly active amorphous aluminosilicate substance, is made by calcining kaolin at 600 to 900 °C for a certain duration [11].

White metakaolin powder, which contains much finer particles than cement (about 3 micrometers), is a very active pozzolan, just like zeolite [6,7]. Since the pozzolanic reaction rate in the concrete containing metakaolin is higher than other pozzolans in the first days, the rates of strength increase and improvement in concrete’s performance in the early days are higher [7].

Furthermore, replacing cement with metakaolin in the concrete mix design reduces porosity and permeability, increases mechanical properties, improves resistance to chemical attacks and alkaline–silica reactions, and finally reduces concrete shrinkage. In view of the abundant and accessible sources of clay soil around the world, the preparation and consumption of metakaolin in construction projects are easy and cost-effective [11].

Based on previous studies, to achieve suitable mechanical and durability properties, the optimum values for replacing cement with metakaolin and zeolite were suggested to be 10–20% in concrete mixtures [1,2,4,7,10,12].

SCMs’ small sizes and specialized surface areas raise hydration heat, reduce concrete workability, and cause cracks in concrete. These cracks can be the entry point for invading ions in corrosive and aggressive environments such as the sea. These cracks correspond to the material type, the amount and type of predicted and unplanned loads, and moisture and heat gradients. Moreover, cracks can be produced by abrasion, erosion, physical and chemical attacks; repeated cycles of wetting and drying; and external forces such as tide forces, ships, winds, and earthquakes. Hence, since cracks reduce the service life of concrete structures in marine conditions, preventing the formation of cracks and decreasing marine concrete crack growth rate is essential. One possible method to reduce the vulnerability of and limit the cracks in concrete is to use fibers. The most significant fibers include synthetic fibers like polypropylene, polyethylene, glass, carbon, and steel fibers as well as natural fibers like cellulose, hemp, and coconut. Moreover, concrete has also incorporated recycled synthetic fibers, like polyethylene terephthalate (PET), and discarded plastic [13].

In cases of initial cracks or deformations, the fibers used in concrete are distributed properly and allow for bridging over the primary cracks, reduce the amount of stress at the cracked edges, restrict the spread and expansion of cracks, and prevent the entry of contaminants and water into the concrete matrix, as a result improving the performance of the concrete [3,13].

On the other hand, using recycled materials from various wastes as fibers in concrete mix design, in addition to improving its various properties, reduces environmental pollution resulting from landfilling or the incineration of waste, air pollution, and soil pollution. Among the wastes in the world, different types of polyethylene, polyamide, or nylon to make plastic food containers or bags are generated more than the other ones. Most of these turn into waste immediately after being used. The waste is then buried in soil or incinerated, which brings about a large amount of pollution and many environmental problems. However, because of the appropriate physical, chemical, and mechanical characteristics of plastic waste, they may safely be utilized as fibers in mixing concrete to increase concrete performance and decrease environmental pollution [14]. The use of SCMs and recycled fibers in concrete mix design has been examined in several studies, some of which are mentioned below.

Yap et al. [15] examined the effect of nylon and polypropylene fibers on the characteristics of oil palm shell concrete. In this study, the workability of concrete decreased when fiber volume increased. On the contrary, the compressive, tensile, and flexural strengths of the concrete exhibited an increase. Adding nylon fibers decreased the modulus of elasticity, but adding polypropylene fibers slightly increased the modulus of elasticity. In a different investigation, Güneyisi et al. [3] demonstrated that lowering the W/B ratio, increasing the number of steel fibers, and substituting metakaolin for 10% of the cement might improve the performance of the material, compressive, tensile, flexural, and bonding strengths. Increasing the length of steel fibers resulted in lower flexural and tensile strengths but higher compressive and bonding strengths. Also, based on scanning electron microscope (SEM) photos, the use of metakaolin and steel fibers improved the interfacial transition zone in concrete. Alyousef et al. [16,17] explored the simultaneous application of recycled fibers from metalized plastic waste fibers and POFA on the properties of green concrete. They found that using these fibers and POFA reduces the compressive strength and concrete workability, while the maxima of the tensile and flexural strengths was obtained by adding 5% fiber and replacing 20% of POFA in concrete. Gautam et al. [18] considered the effects of solid waste like ceramic waste as SCMs and aggregates on the physical, mechanical, and durability properties of concrete mixtures. They revealed that adding ceramic waste reduces the cost of construction material and emission of global greenhouse gas, helps in attaining sustainability, and improves mechanical and durability properties of concrete. Alani et al. [19] found that the compressive strength, porosity, and chloride permeability of concrete specimens were increased compared to plain concrete by adding SF from PET bottle waste to ultra-high-performance green concrete containing POFA. Al-Darzi [20] studied the effects of recycled plastic waste (RPET) in concrete slabs. They showed that by using RPET in concrete mixes, the density, compressive, and splitting tensile strengths were decreased. Moreover, by replacing 5% of the control slab’s material with RPET, crack width, ultimate load, and early loading stage deflection were decreased. Dong et al. [21] presented a sustainable green self-compacting concrete by using waste plastic, industry by-products (fly ash, slag, and silica fume), and waste concrete aggregate. By increasing RPF content, the slump flow diameters decreased and T slump flow times and SF-JF values increased, but the segregation resistance and compressive strength were improved. Also, the addition of SCMs and RPFs enhanced the elastic modulus. In addition, the corporation of RPFs significantly increased flexural strength and toughness. Gautam et al. [22] considered the feasibility of bone china ceramic powder waste (BCPW) and granite waste (GW) as a cement and fine aggregate in SCC mixes. The maximum strength was found for SCC mixes containing 10% BCPW and 30% GW. The optimum mixes obtained higher density, less permeable voids, increased carbonation resistance, help in cost reduction, and also offered sustainability.

Thus, according to previous research, it was suggested that 0.5–1% of the concrete volume use recycled plastic fibers with 10–20% SCMs to obtain better performance for concrete mixes [15,16,19,21,23,24,25].

Considering this background, in the maritime environment, little investigation has been conducted into green concrete flexural behavior caused by recycled fibers and metakaolin/zeolite. Recycled fiber used alongside metakaolin/zeolite as an SCM seems to decrease environmental pollution and maintain natural resources, leading to sustainable development. Moreover, with this kind of green concrete, it will be feasible to increase the flexural strength, ductility, and durability of maritime concrete structures. On the other hand, by using fibers, cracking in marine concrete structures is limited and their damage and failure rate is reduced. The current research program investigated the effect of recycled fibers like disposable glass (RF/SF forms) on flexural capacity, and the toughness of green concrete beams containing metakaolin/zeolite was examined. Furthermore, the microstructures of concrete mix designs were investigated.

## 2. Experimental Program

### 2.1. Materials

#### 2.1.1. Cement

In the current research, for preparing concrete mixes, Portland cement Type II from the Kerman cement factory was used. Table 1 lists the cement’s chemical and physical properties.

#### 2.1.2. Metakaolin

Metakaolin from Tehran, Iran was used to prepare the concrete mixtures. Metakaolin’s chemical and physical characteristics are shown in Table 1. Metakaolin particles that passed through a #200 sieve (particles size < 0.075 mm) were used to prepare the concrete mixes. The particle size of metakaolin is finer than cement, but the specific gravity of metakaolin is lower than cement. Furthermore, metakaolin has higher specific surface area than cement. Metakaolin was added to the concrete mix in quantities of 10 and 20% by cement weight.

#### 2.1.3. Zeolite

The zeolite used in this study was taken from Semnan, Iran. Zeolite’s chemical and physical characteristics are shown in Table 1. Similarly to metakaolin, zeolite particles that passed through a #200 sieve (particles size < 0.075 mm) were used to prepare the concrete mixes. The particle size of zeolite is finer than cement, while the specific gravity of cement is higher than zeolite. In addition, the specific surface area of cement and zeolite is almost the same. Zeolite was added in proportions between 10 and 20% of the cement weight in the concrete mixes.

#### 2.1.4. Aggregates

Natural river sand with a maximum size of 4.75 mm (passing a #4 sieve) was utilized as a fine aggregate, while crushed gravel with a maximum size of 19 mm was coarse aggregate. Some aggregate characteristics are presented in Table 2. The source of all aggregates used in this study was mines located in Chabahar, Iran. Also, ASTM C136 was used to analyze aggregates using sieves [29].

#### 2.1.5. Water

In this study, drinking water was utilized to mix all of the concrete and to cure the specimens in the tanks at early ages. The drinking water was extracted, purified, and distributed at Chabahar Maritime University and controlled at the water quality control laboratory.

#### 2.1.6. Superplasticizers

A superplasticizer is one of the essential components in concrete mixes to enhance fiber-reinforced concrete workability. In the current research, a type of superplasticizer based on modified polycarboxylates called Farco Plast (From Shimi Sakhteman factory, Tehran, Iran) was used. Farco Plast superplasticizer is suitable for concreting in hot weather conditions, concretes with long transportation time, concretes with very low slump, and concretes containing fibers and SCMs. It has a specific gravity of around 1.09 kg/L and is light brown. Also, after mixing with water, superplasticizer is added to the concrete mixture for controlling the workability of mixtures.

#### 2.1.7. Recycled Fibers

The polyethylene recycled fibers used in this study were divided into two main groups: ring and strip disposable glass fibers. The used disposable glasses were cleaned and washed after being collected from public places such as the beach, schools, malls, stores, restaurants, hotels, etc. Then, the disposable glasses were cut into string and ring shapes by hand with scissors, sorted into specific sizes, and weighed. Table 3 lists the fiber characteristics for concrete mix design. Based on Table 3, the aspect ratio of fibers obtained by dividing the length of the fibers by their diameter were 40 and 80. Formula is:(1)$$\lambda =\frac{l}{d_{e}}=\frac{l}{2\times \sqrt{\frac{A}{\pi }}}=\frac{l}{2\times \sqrt{\frac{b\times c}{\pi }}}$$
where *λ* is aspect ratio, *l* is the fiber length (mm), *d_e_* is the equivalent diameter (mm), *A* is the fiber cross-section area (mm^2^), *b* is fiber width (mm), and *c* is fiber thickness (mm) [23,30].

##### Disposable Glass RF

By horizontally slicing disposable glasses, polyethylene fibers used to make the ring of the glass are produced (closed loops). These rings have variable widths equal to 5 and 10 mm, and their diameter is about 60 ± 5 mm. In addition, the volume fraction of this type of fiber in the concrete mix design is equal to 0.5 and 1% of the concrete volume.

##### Disposable Glass SF

Disposable glasses were sliced vertically or horizontally to create polyethylene SFs (strip). These strips have a fixed width of 10 mm and varying lengths of 25 and 50 mm. Meanwhile, the volume fraction of this type of fiber in the concrete mix design was set as 0.5 and 1% of the concrete volume. All fiber types are shown in the Figure 1.

### 2.2. Mixture Properties

In this study, 36 green concrete mix designs with different amounts of cement replacements including metakaolin, zeolite, and recycled fibers such as RF and SF from disposable glasses were prepared. Besides these specimens, concrete specimens without any fibers were also prepared as control specimens. In all mix designs, the water/binders was equal to 0.5. Furthermore, the total weight of cementitious materials in all mixtures was 410 kg/m^3^. Farco Plast superplasticizer was also employed to increase the concrete’s workability by 2% by cement weight due to the usage of SCMs and fibers in the concrete mixes. Table 4 shows the characteristics of the concrete mixtures.

### 2.3. Specimen Preparation

Depending on the type of test, all the specimens had to be left in the molds for 24 h after their preparation. After being removed from the molds, the specimens were cured in a freshwater tank for 3 days outside the laboratory (Figure 2) and finally moved to the Oman Sea’s tidal zone (Figure 3).

## 3. Exposure Conditions

Since most deterioration happens in marine concrete structures in the tidal zone, the environment of the study was the Oman Sea’s tidal zone. Therefore, all the specimens were placed on the Oman Sea shore. Table 5 shows the chemical parameters of Oman Sea water and concrete water. Furthermore, the tidal zone weather conditions (Chabahar Gulf in Oman Sea) at experimental program time are presented in Table 6.

## 4. Testing Methods

### Flexural Behavior

The flexural behavior was investigated by determining the maximum load capacity (P_max_) and flexural toughness of several green concrete beams. This test was performed according to ASTM C78 [31] and ASTM C1609 [32] standards for plain and fiber-reinforced concrete specimens with dimensions of 350 × 100 × 100 mm. In addition, the loading speed of the device was considered constant and equal to 0.5 (mm/min). At the ages of 28, 90, and 180 days, the specimens that were in the Oman Sea tidal zone were tested for flexural strength. The maximum deflection was set at 10 mm for all the beams to allow us to evaluate the flexural toughness.

## 5. Results and Discussion

### 5.1. Flexural Behavior

Table 7 and Table 8 show the effect of the RF and SF on maximum load capacity and flexural toughness of metakaolin/zeolite fiber-reinforced concrete beams at 28, 90, and 180 days.

#### 5.1.1. Flexural Behavior of RF Concrete

In Figure 4, Figure 5, Figure 6, Figure 7, Figure 8 and Figure 9, the flexural behavior of RF concrete beams with zeolite/metakaolin at 28, 90, and 180 days is displayed.

According to Figure 4, Figure 5, Figure 6, Figure 7, Figure 8 and Figure 9, the concrete containing 10 or 20% metakaolin with the increase in fiber volume P_max_ decreased at all ages. The largest amount of decrease was related to the 5 mm fiber width. P_max_ was reduced by 21 and 27% in the concrete mix with 10% metakaolin at 28 days when 0.5 and 1% RF were added, respectively. Moreover, at 90 days, concrete beams with 0.5 and 1% RF had 18 and 14% lower P_max_ than plain concrete, respectively. In addition, at 180 days, 0.5 and 1% RF reduced the P_max_ of concrete beams by 27 and 25%, respectively.

On the other hand, in the concrete mix with 20% metakaolin, adding 0.5 and 1% RF at 28 days lowered P_max_ by 21 and 30%, respectively. In addition, the P_max_ of green concrete with 0.5 and 1% RF was 20 and 32% lower than without fibers at 90 days. Finally, after 180 days, the green concrete with 20% metakaolin observed a P_max_ reduction of 17 and 31%, respectively, when RF was added at amounts of 0.5 and 1%.

Based on Figure 4, Figure 5, Figure 6, Figure 7, Figure 8 and Figure 9, the P_max_ was reduced with increasing fiber volume in concrete with 10 or 20% zeolite at all ages. Adding 0.5 and 1% RF to concrete with 10% zeolite at 28 days decreased P_max_ by 20 and 28%, respectively. On the other hand, at 90 days, concrete with 0.5 and 1% RF had 15 and 24% lower P_max_ values than green concrete without fibers. Additionally, after 180 days, the P_max_ was reduced by 9 and 21%, respectively, when RF 0.5 and 1% were added to the green concrete. Moreover, in 20% zeolite concrete at 28 days, by adding 0.5 and 1% RF, the P_max_ decreased up to 22 and 30%, respectively. Furthermore, on the 90th day, the P_max_ values of concrete with 0.5 and 1% RF were lower than green concrete without fibers by 18 and 28%, respectively. In addition, at 180 days, adding 0.5 and 1% RF to the green concrete with 20% zeolite led to decreases of 13 and 25% in P_max_, respectively.

Because the characteristics of the cement matrix are one of the biggest and most effective variables used to determine the P_max_, incorporating fibers into the concrete causes changes in the characteristics of the cement matrix. In fact, in the early days, fibers enhanced concrete specimens’ empty areas and porosity; resultantly, their P_max_ decreased. After many years, due to the progress of the hydration process and the use of metakaolin/zeolite, which are usually finer than cement, the concrete density increased. As a result, the difference between the P_max_ of beams with/without fibers is decreased [5,23].

Further, based on Figure 4, Figure 5, Figure 6, Figure 7, Figure 8 and Figure 9, in mixtures at 28 days, in comparison to specimens containing zeolite, those containing metakaolin showed greater P_max_ values. In early time stages, metakaolin’s finer size and higher pozzolanic activity than zeolite accelerate hydration processes, increase CSH gel synthesis, reduce porosity, and enhance the fiber and cement paste transition zone. However, as time goes on, the growth rate of P_max_ in the specimens containing zeolite is higher than that for specimens containing metakaolin [24,25,26].

According to Figure 4, Figure 5, Figure 6, Figure 7, Figure 8 and Figure 9, by incorporating fibers into green beams, their flexural toughness increased more noticeably. For instance, at 28 days, incorporating 0.5% RF with widths of 5 and 10 mm to the green concrete caused increases in the flexural toughness by 857 and 1012%, respectively. In addition, at 90 days, incorporating 0.5% RF with widths of 5 and 10 mm to the plain green concrete caused an enhancement in the toughness of green beams by 914 and 1055%, respectively. On the other hand, after 180 days, beam flexural toughness with 0.5% RF with widths of 5 and 10 mm caused increases to 980 and 1137%, respectively, compared to plain green concrete beams. Furthermore, at 28 days, adding 1% RF with widths of 5 and 10 mm to the simple green concrete caused improvement in flexural toughness up to 1096 and 1278%, respectively. Meanwhile, at 90 days, the flexural toughness of beams with 1% RF with widths of 5 and 10 mm increased by 1199 and 1380%, respectively, when compared with plain green concrete beams. On the other hand, incorporating 1% RF with widths of 5 and 10 mm to the observed green concrete caused an increase in flexural toughness up to 1341 and 1582% at 180 days, respectively [33,34].

Furthermore, by changing the volume fraction of RF from 0.5 to 1%, green fiber-reinforced concrete beams’ flexural toughness increased. To be more exact, raising the volume fraction of RF from 0.5 to 1% enhanced flexural toughness by 34, 38, and 43% at 28, 90, and 180 days, respectively. Since beams without fibers start to crack and break suddenly under loading, the presence of fibers delays the occurrence of cracks, reduces the growth rate of possible cracks, and increases energy absorption (toughness). In general, the behavior mechanism of fibers in crack control is known as crack bridging, which is the main factor in improving the toughness and ductility of concrete sections. It appears that by raising the fibers’ volume fraction, the crack bridging increases, which leads to a greater distribution of the applied force by the fibers, more energy absorption, and an increase in the toughness of the concrete beams [23,24,34].

Based on Figure 4, Figure 5, Figure 6, Figure 7, Figure 8 and Figure 9, it is clear that in the early stages, using metakaolin in green fiber-reinforced concrete beams caused greater increase in toughness compared with a similar mixture containing zeolite. As an example, the maximum obtained toughness in metakaolin/zeolite fiber-reinforced concrete beams increased by 1278 and 1207%, respectively. As time went on, the effects of zeolite on green fiber-reinforced concrete beams’ flexural toughness became higher than metakaolin, such that after 180 days, the maximum toughness increases in green fiber-reinforced beams containing zeolite and metakaolin were 1582 and 1437%, respectively [35,36].

In green fiber concretes containing SCM, by increasing the amount of SCM from 10 to 20%, the toughness of green fiber-reinforced concrete beams decreased. For example, in green fiber beams with 1% RF, with the increase in metakaolin dosage at 28 and 180 days, the toughness decreased by 13 and 15%. Also, by increasing the percentage of zeolite in green fiber-reinforced concrete, the flexural toughness of the beam decreased so that after 28 and 180 days, the beams’ toughness decreased by 9 and 13%, respectively. Due to fine particle size, filling properties, and pozzolanic reaction, metakaolin and zeolite used in concrete strengthen the transition zone, reduce the number and volume of holes, increase the density of concrete, and eventually, improve cement paste–fiber adhesion. In the early ages, the effect of metakaolin on concrete properties is greater than zeolite. However, after a long time, zeolite, by production of CSH secondary gels, causes improvement in microstructure, stronger bonds between the fiber–cement matrix, a greater decrease in porosity, and a further increase in flexural toughness when compared to metakaolin [5,37,38,39,40].

#### 5.1.2. Flexural Behavior of SF Concrete

In Figure 10, Figure 11, Figure 12, Figure 13, Figure 14 and Figure 15, the flexural behavior of SF concrete beams with zeolite/metakaolin at 28, 90, and 180 days is shown.

Figure 10, Figure 11, Figure 12, Figure 13, Figure 14 and Figure 15 show that, in a concrete mixture with 10 or 20% metakaolin, by increasing the SF volume, the P_max_ decreases at different ages. For instance, at 28 days, in concrete mixtures containing 10% metakaolin, the P_max_ of concrete reinforced with 0.5 and 1% SF decreased by 27 and 33%, respectively, relative to green concrete without fibers. But at 90 days, with the addition of 0.5 and 1% of SF, the P_max_ decreased to 23 and 32%, respectively. Moreover, at 180 days, the addition of 0.5 and 1% of SF caused a decrease in P_max_ of plain green concrete by 25 and 34%, respectively. Furthermore, in concrete beams containing 20% metakaolin at 28 days, the addition of 0.5 and 1% SF caused a decrease in P_max_ by 29 and 38%, respectively. In addition, after 90 and 180 days, compared to green concrete without fiber, the P_max_ of concrete containing 0.5 and 1% of SF reduced by 26 and 37%, respectively.

Based on Figure 10, Figure 11, Figure 12, Figure 13, Figure 14 and Figure 15, by raising concrete mixes’ SF volume fraction with 10 or 20% zeolite at different ages, the P_max_ decreased. At 28 days, incorporating 0.5 and 1% of SF into concrete containing 10% zeolite led to decreases in the P_max_ values of plain green concrete by 25 and 35%, respectively. In addition, at 90 days, the P_max_ values of concrete with 0.5 and 1% of SF decreased by 22 and 32%, respectively. On the other hand, with the incorporation of 0.5 and 1% of SF at 180 days, the P_max_ values decreased by 18 and 28%, respectively. In addition, in concrete containing 20% zeolite at 28 days, with the addition of 0.5 and 1% SF, the P_max_ reduced by 29 and 37%, respectively. Furthermore, at 90 days, the addition of 0.5 and 1% SF caused decreases of 27 and 35% in P_max_, respectively. Finally, after 180 days, green concrete without fibers had a P_max_ 23% lower than concrete with 20% zeolite and 0.5 and 1% SF. It is apparent that by incorporating fibers into plain concrete, cohesion decreases, the cement matrix becomes weak and porous, and finally P_max_ decreases. In addition, by increasing the fiber volume, the P_max_ decreases further [24].

Figure 10, Figure 11, Figure 12, Figure 13, Figure 14 and Figure 15 show that the P_max_ of green fiber-reinforced concrete beams increases by raising the aspect ratio of the fibers. For example, S4 with 80 aspect ratio had 14% greater P_max_ than S3 with 40 aspect ratio, which contains 0.5% fibers, on all days. However, S12 with 80 aspect ratio had a 13% higher P_max_ than mixes with 40 aspect ratio in green concrete beams with 1% fibers. Therefore, by raising the fiber’s aspect ratio, the development length of the fibers in the cement matrix, the resistance to the applied force (P_max_), and the resistance to cracking of the section are all increased [30].

Furthermore, comparing Figure 10, Figure 11, Figure 12, Figure 13, Figure 14 and Figure 15 reveals that, like concrete specimens containing RF, the SF-reinforced concrete samples containing zeolite had lower P_max_ values compared to concrete containing metakaolin in the short term. However, for long periods, the P_max_ of samples containing zeolite was slightly higher than that of samples containing metakaolin [36].

According to Figure 10, Figure 11, Figure 12, Figure 13, Figure 14 and Figure 15, fibers significantly improved green concrete beams’ flexural toughness. For example, by incorporating 0.5% of SF with 40 and 80 aspect ratios, the flexural toughness of the green concrete without fibers increased by 992 and 1187% at 28 days, respectively. In addition, at 90 days, incorporating 0.5% of SF with 40 and 80 aspect ratios to green concrete without fibers caused increases in the toughness of plain green beams by 1039 and 1234%, respectively. In addition, the flexural toughness of beams with 0.5% SF with 40 and 80 aspect ratios increased by 1154 and 1406% at 180 days, respectively. Furthermore, at 28 days, incorporating 1% SF with 40 and 80 aspect ratios to the green concrete without fibers caused increases in the flexural toughness by 1323 and 1561%, respectively. Meanwhile, at 90 days, the flexural toughness of beams with 1% SF with 40 and 80 aspect ratios increased by 1417 and 1680%, respectively. Also, after 180 days, the toughness of green concrete beams increased by 1609 and 1975% by adding 40 and 80 aspect ratios of fibers, respectively.

Figure 10, Figure 11, Figure 12, Figure 13, Figure 14 and Figure 15 showed that the toughness of beams increased with the aspect ratio of fibers. For example, in green fiber-reinforced concrete beams with 0.5% SF, the flexural toughness of S4 compared to S3, at different days, increased by up to 22%. Moreover, in mixtures with 1% SF, the flexural toughness of S12 compared to S11 increased by 17 to 19% up to 180 days. Due to the increased fiber aspect ratio, the contact surface and the adhesion between the fibers–cement paste was enhanced. On the other hand, by raising the length of the fibers, the fibers’ slipping and pullout resistance increased [30].

According to Figure 10, Figure 11, Figure 12, Figure 13, Figure 14 and Figure 15, it is clear that in concrete beams with SF at 28 days, beams containing zeolite had lower flexural toughness than beams containing metakaolin. However, the flexural toughness of beams with zeolite was higher than that of metakaolin concrete beams after 180 days. Based on Figure 10, Figure 11, Figure 12, Figure 13, Figure 14 and Figure 15, at 28 days, it was clear that adding metakaolin to fiber-reinforced concrete beams caused a greater increase in flexural toughness than in zeolite ones. In other words, at 28 days, the maximum increases in green fiber-reinforced concrete beams’ flexural toughness with metakaolin and zeolite were 1561 and 1454% higher, respectively. Moreover, at 180 days, the toughness values of beams with zeolite or metakaolin increased by up to 1975 and 1797%, respectively. Therefore, for long periods, zeolite increased the toughness of beams more than metakaolin [35,36,41].

On the other hand, according to Figure 10, Figure 11, Figure 12, Figure 13, Figure 14 and Figure 15, the toughness of beams with 10% SCMs was higher than that of beams with 20% SCMs. For example, by increasing the amount of metakaolin from 10 to 20% in beams with 1% SF, the flexural toughness decreased by up to 14% at different ages. Moreover, in green concrete mixtures with 1% SF, with increasing the amount of zeolite from 10 to 20%, the flexural toughness of the beams decreased by up to 15% until 180 days. Although the use of metakaolin and zeolite as SCMs might improve the properties of fiber-reinforced concrete, replacing the higher percentage of cement with metakaolin can cause reduce the concrete workability, adhesion, and strength of the matrix. Moreover, due to a high specific surface area and the three-dimensional and porous structure of zeolite, a considerable amount of mixed water is absorbed by zeolite such that the homogeneity of the concrete is reduced. Therefore, the homogeneity, integrity, and toughness decrease [5,37,38,39,40].

The green concrete beams’ flexural behavior with RF and SF on all days is shown in Figure 5, Figure 6, Figure 7, Figure 8, Figure 9, Figure 10, Figure 11, Figure 12, Figure 13, Figure 14, Figure 15 and Figure 16. Figure 5, Figure 6, Figure 7, Figure 8, Figure 9, Figure 10, Figure 11, Figure 12, Figure 13, Figure 14, Figure 15 and Figure 16 show that P_max_ in green concrete beams with RF was higher than that of the mixture with SF. Furthermore, the P_max_ of green concrete beams with 0.5% fibers such as R1 was up to 12% higher than S1 at different ages. However, the P_max_ for R16, which contains 1% fiber, was at maximum 11% higher than S16. Due to the bigger size of RF compared to SF, the P_max_ in beams with RF may be higher than that of the beams containing SF. This could be because of an increase in RF developmental length in the cement matrix and the adhesion of fibers to concrete [42,43].

The comparison of Figure 4, Figure 5, Figure 6, Figure 7, Figure 8, Figure 9, Figure 10, Figure 11, Figure 12, Figure 13, Figure 14 and Figure 15 reveals that in fiber green concrete with the same characteristics, the flexural toughness of beams with SF recycled from disposable glass was higher than that of the beams with RF. For example, in mixtures containing 0.5% fibers, the flexural toughness of S1 compared to R1 increased by up to 16% until 180 days. Additionally, in concrete beams containing 1% fibers, S16, compared to R16, the flexural toughness increased by a maximum of 18% at different ages. Therefore, it seems that the difference between the toughness of green concrete beams with RF and SF increases slightly by raising the fiber’s volume fraction in concrete. Since, in a constant volume of concrete, the amount of SF is greater than that of RF, it seems that more force is needed for the initiation and development of cracks. In addition, using SF, which is smaller than RF, produces less porosity but improves the quality of the transition zone between fibers–cement pastes. Therefore, the toughness and ductility of concrete beams with SF will be higher than those of RF-reinforced concrete [25,44,45].

### 5.2. Scanning Electron Microscope (SEM)

SEM may be used to study cement paste morphology and the interface transition zone (ITZ) between cement pastes and aggregates/fibers. Furthermore, SEM can also evaluate cement paste compactness, pore size and shape, hydration product structural morphology, and ITZ quality between cement pastes and aggregates/fibers.

To prepare the SEM samples, 7 cm concrete cubes with different mixes were used. Then, small pieces of the concrete samples were covered with gold sheets to take better-quality photos. The photos were prepared with different magnifications to better evaluate the characteristics of the concrete samples.

A dense and compact microstructure was created by the joining of the SCM particles and the hydration products. CSH gels from the pozzolanic reaction of SCMs tightly wrapped the hydration products and solute particles, forming a dense continuous phase. The CSH gels bonded the aggregate and cement mortar and filled interior voids and cracks, decreasing porosity and enhancing microstructure. Metakaolin/zeolite with different particle sizes may fill voids and cracks simultaneously. According to the cement paste SEM images, metakaolin/zeolite improved the cement mortar microstructure and mechanical properties of the fiber concrete samples.

On the other hand, ITZ between cement pastes and aggregates consists of aggregate, cement mortar, cracks, pores, and unhydrated particles. The incorporation of metakaolin/zeolite SCMs significantly improved ITZ microstructure. The ITZ between aggregates and cement paste had much smaller pores and cracks. Thus, the addition of metakaolin/zeolite enhanced the mechanical properties. The ITZ’s Ca(OH) crystal enrichment was reduced by the pozzolanic reaction of metakaolin/zeolite, which consumed many Ca(OH)_2_ crystals. The hydration products, such as CSH gels, produced by the pozzolanic reaction of SCMs can fill ITZ cracks and pores, improving mechanical interlocking and microstructure and improving concrete mix mechanical properties.

Fibers are embedded in the cement paste and surrounded by hydration products. This reduces separation and settlement cracks and synergistically improves bridge cracks, making fiber concrete mixtures stronger and tougher. The unreacted SCM particles and hydration products filled holes and cracks, increased ITZ compactness, enhanced the bond strength of the fibers and cement paste, and improved the compressive, flexural, and pull-out strength. A SEM investigation confirmed the fiber–SCM coupling’s good performance. SCMs and fibers coupled had a better coupling effect, which increased compressive and flexural strength more than either element on their own [21].

In the following, to better evaluate the microstructural characteristics of green concrete containing recycled plastic fibers and metakaolin/zeolite, SEM images are shown in Figure 17. Also, the contact area between fibers and cement paste, which is very impactful on the properties and quality of concrete, is presented in Figure 18. Moreover, cement hydration products like C-S-H gels, HC, and ettringite are illustrated in Figure 19. In addition, in Figure 20, cement paste details, micro/macro cracks, and pores are shown.

Eventually, adding SCMs to the fiber concrete mixtures, due to high pozzolanic activity, causes the production of more CSH gels, improves and strengthens the concrete microstructure, increase the quality of the cement matrix and bond strength, and reduces porosity and cracks at the contact surface of fibers and the cement matrix. Furthermore, by increasing the adhesion strength and pull-out of fibers from the cement matrix, the flexural strength and toughness increased.

Finally, the results show that using fibers made from recycled disposable glasses in the form of strips and rings composed of metakaolin/zeolite as SCMs has an acceptable performance in reducing marine environmental pollution, improving ductility, and controlling cracks in marine concrete structures.

## 6. Conclusions

In this paper, the impacts of using 0.5 and 1% disposable glass RF and SF on green concrete beams’ flexural behavior with 10 or 20% metakaolin and zeolite as SCMs at 28, 90, and 180 days in the Oman Sea’s tidal zone were investigated. The findings suggest the following results:The maximum load capacities of green concrete beams with RF/SF were decreased up to 31 and 37% compared to control specimens, respectively. Although in the short term, the maximum load capacity for metakaolin concrete was greater than zeolite concrete, the results were reversed in the long term.Green concrete beams containing RF had maximum load capacity up to 13% greater than green concrete beams containing SF.The flexural toughness values of green concrete with 0.5 and 1% RF were increased up to 11 and 16 times more than the control specimens, respectively.The flexural toughness values of 0.5 and 1% SF-reinforced concrete containing metakaolin and zeolite were increased up to 14 and 20 times more than the control specimens, respectively.The flexural toughness of green concrete beams with SF was 24% greater than green concrete beams containing RF.In the short term, the flexural toughness of beams containing metakaolin increased by up to 13% than zeolite concrete beams. But, for long periods, beams with zeolite exhibited 8% greater flexural toughness than those with metakaolin.The evaluation of the SEM showed that although the use of microplastic fibers increased the porosity of concrete, adding SCMs (metakaolin/zeolite) to concrete mixes greatly reduced the porosity and its negative effects. Also, the fiber–SCM coupling performance was better than those of the separate elements.By using SCMs (metakaolin/zeolite) and recycled disposable glass fibers in concrete mixtures, air and marine pollution decrease, ductility increases, and cracks are limited; thus, the durability and service life of marine concrete structures are enhanced.

## Figures and Tables

**Figure 1 materials-16-05912-f001:**
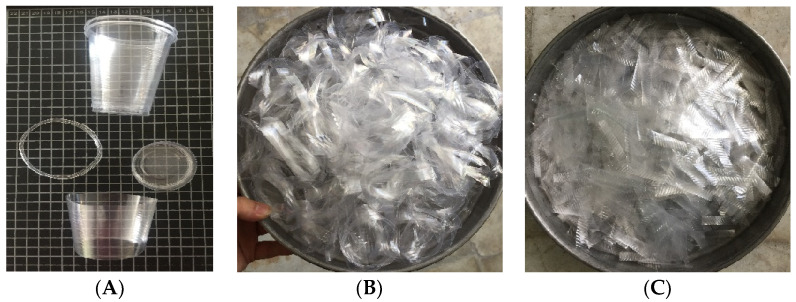
Fibers: (**A**) making disposable glass fibers, (**B**) RF, (**C**) SF.

**Figure 2 materials-16-05912-f002:**
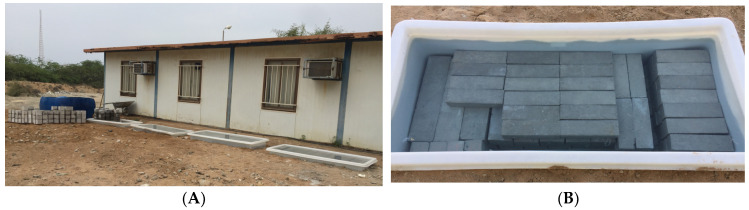
(**A**) Curing area of concrete samples (outside the laboratory), (**B**) freshwater curing tank.

**Figure 3 materials-16-05912-f003:**
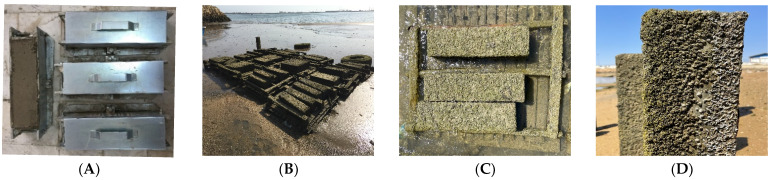
(**A**) Specimen molds, (**B**) Oman Sea tidal zone, (**C**) Tidal zone samples, (**D**) Specimens after leaving the tidal zone.

**Figure 4 materials-16-05912-f004:**
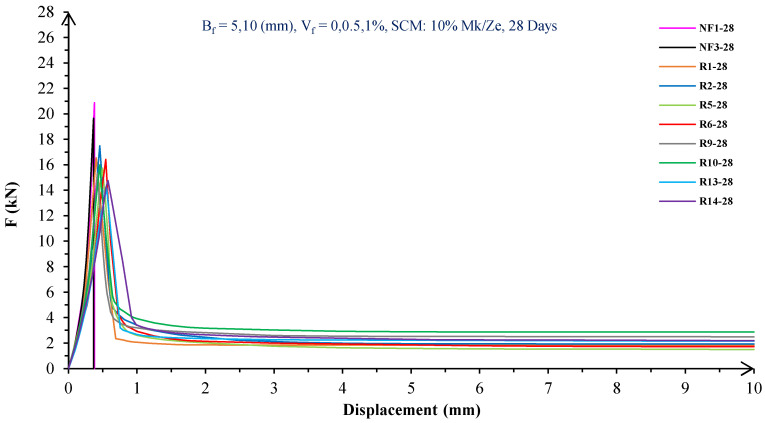
Flexural behavior of RF concrete beams with 10% metakaolin/zeolite (28 days).

**Figure 5 materials-16-05912-f005:**
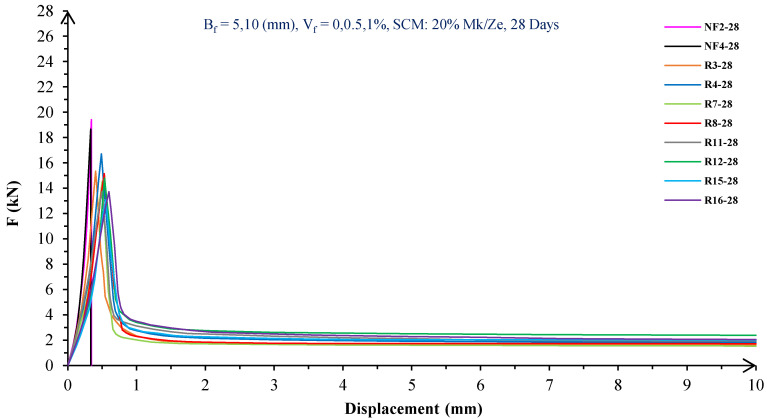
Flexural behavior of RF concrete beams with 20% metakaolin/zeolite (28 days).

**Figure 6 materials-16-05912-f006:**
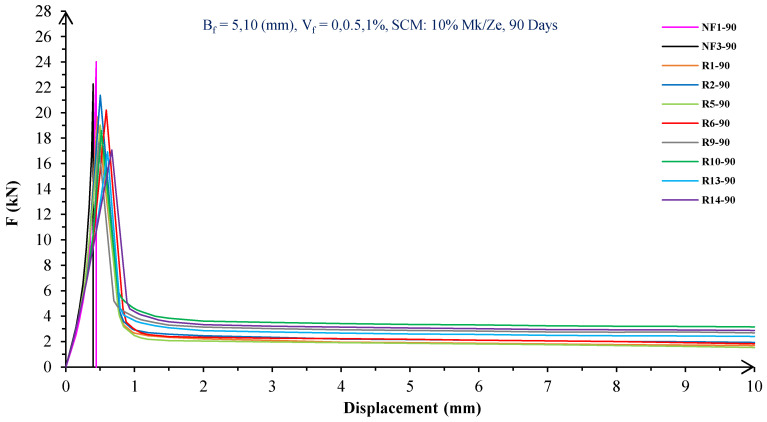
Flexural behavior of RF concrete beams with 10% metakaolin/zeolite (90 days).

**Figure 7 materials-16-05912-f007:**
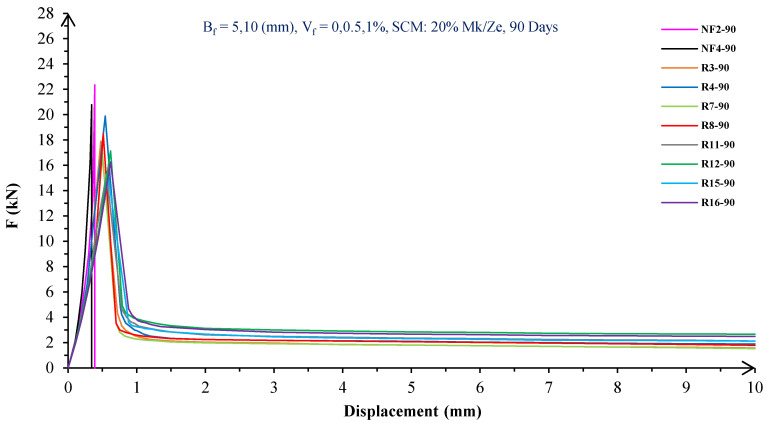
Flexural behavior of RF concrete beams with 20% metakaolin/zeolite (90 days).

**Figure 8 materials-16-05912-f008:**
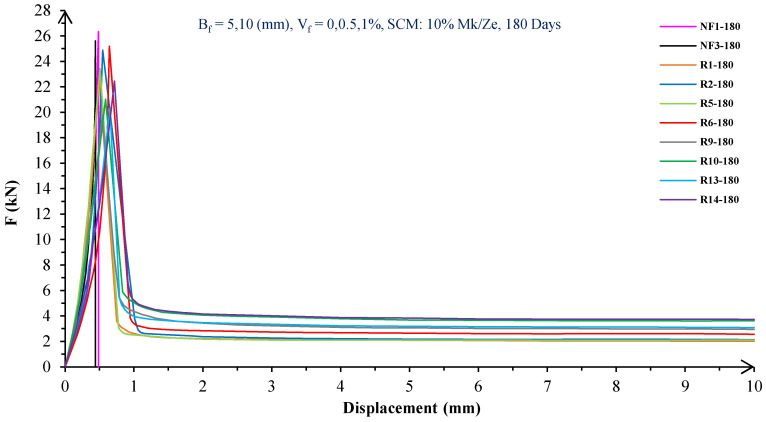
Flexural behavior of RF concrete beams with 10% metakaolin/zeolite (180 days).

**Figure 9 materials-16-05912-f009:**
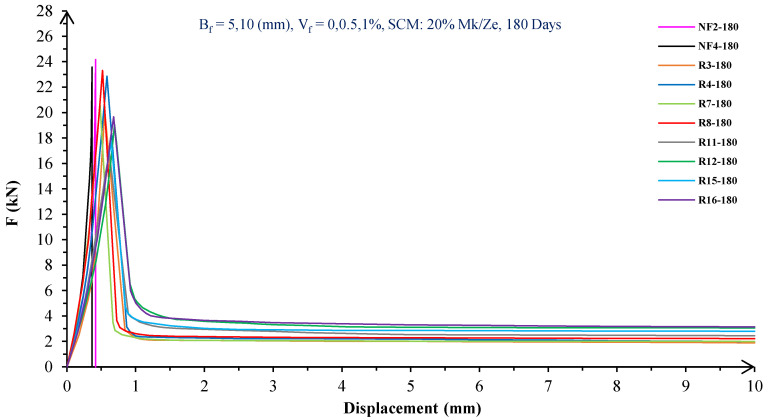
Flexural behavior of RF concrete beams with 20% metakaolin/zeolite (180 days).

**Figure 10 materials-16-05912-f010:**
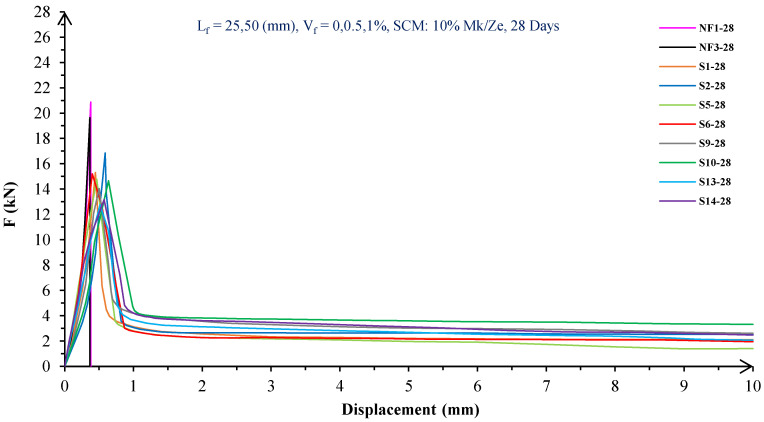
Flexural behavior of SF concrete beams with 10% metakaolin/zeolite (28 days).

**Figure 11 materials-16-05912-f011:**
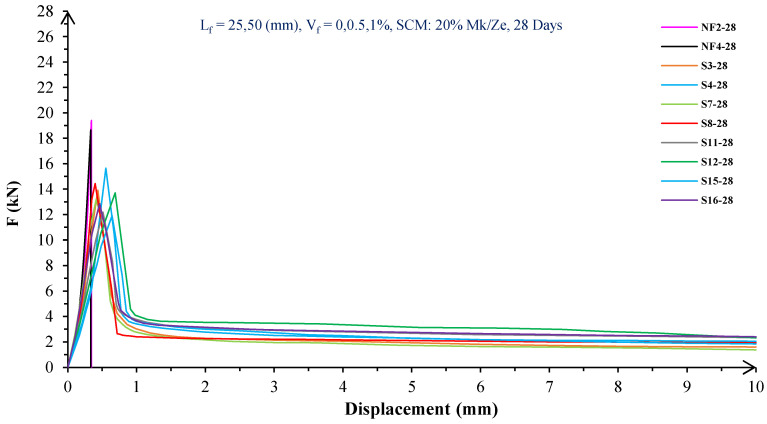
Flexural behavior of SF concrete beams with 20% metakaolin/zeolite (28 days).

**Figure 12 materials-16-05912-f012:**
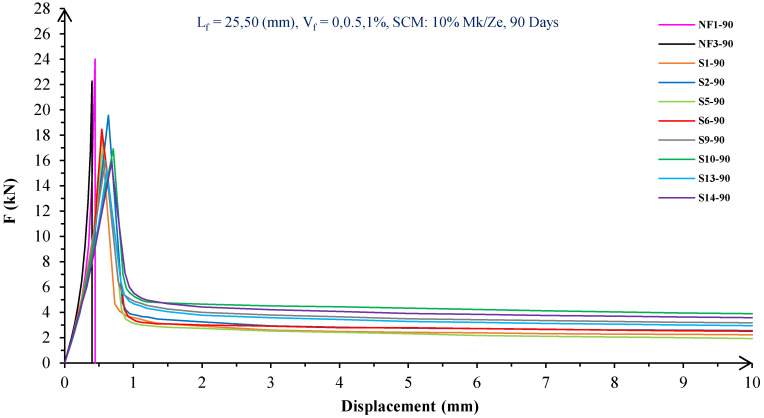
Flexural behavior of SF concrete beams with 10% metakaolin/zeolite (90 days).

**Figure 13 materials-16-05912-f013:**
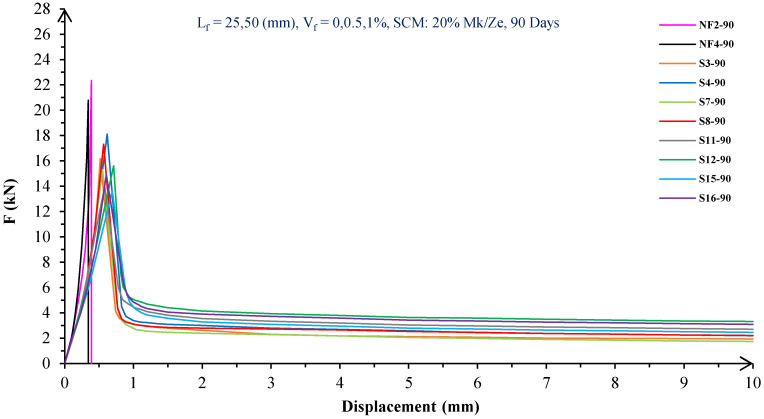
Flexural behavior of SF concrete beams with 20% metakaolin/zeolite (90 days).

**Figure 14 materials-16-05912-f014:**
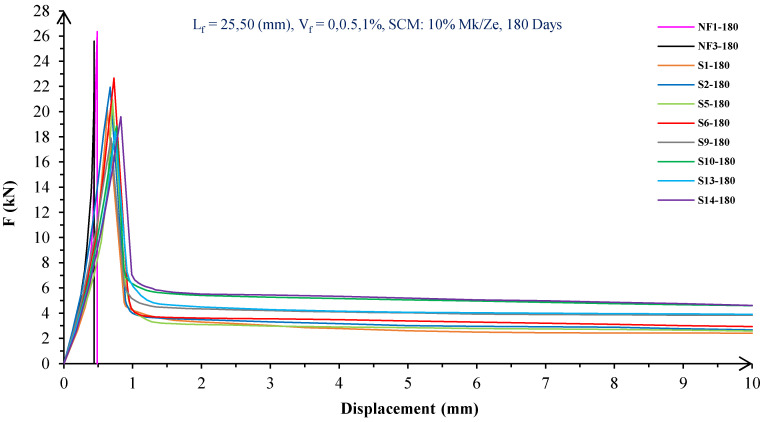
Flexural behavior of SF concrete beams with 10% metakaolin/zeolite (180 days).

**Figure 15 materials-16-05912-f015:**
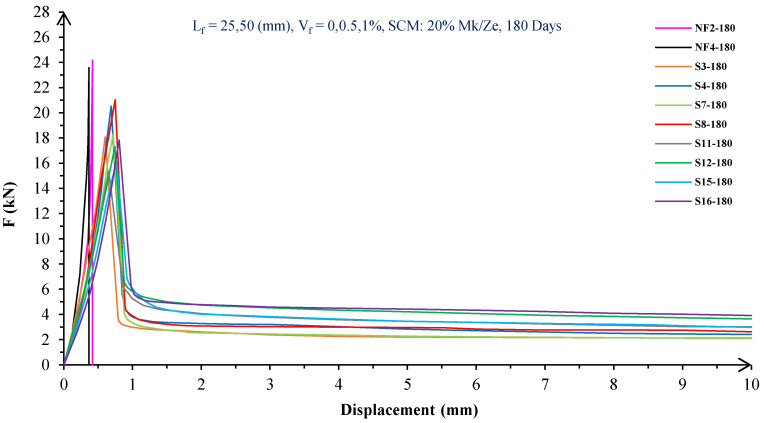
Flexural behavior of SF concrete beams with 20% metakaolin/zeolite (180 days).

**Figure 16 materials-16-05912-f016:**
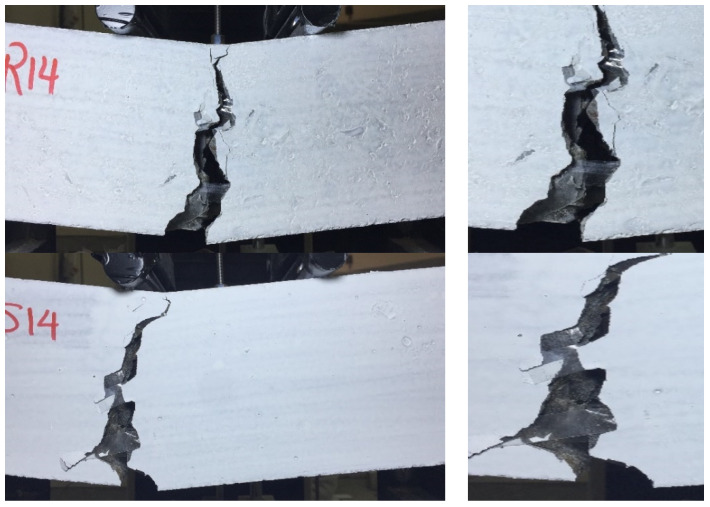
Flexural tests for concrete samples containing RF and SF (R14 and S14).

**Figure 17 materials-16-05912-f017:**
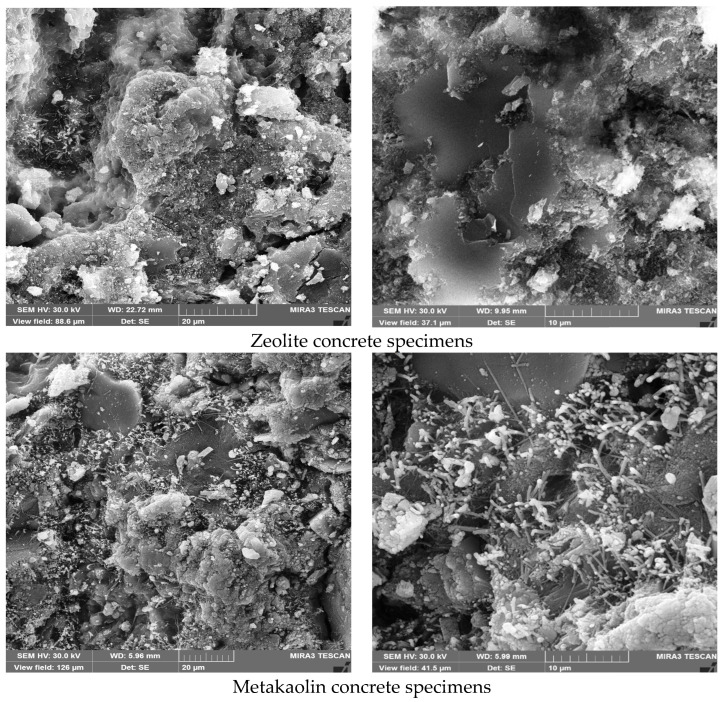
The microstructural properties of metakaolin/zeolite concrete.

**Figure 18 materials-16-05912-f018:**
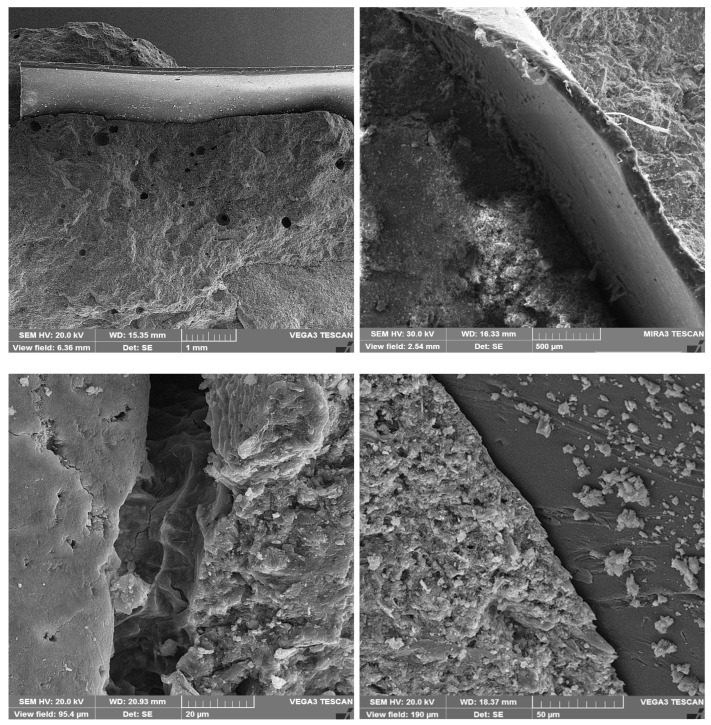
The fiber–cement paste contact zone.

**Figure 19 materials-16-05912-f019:**
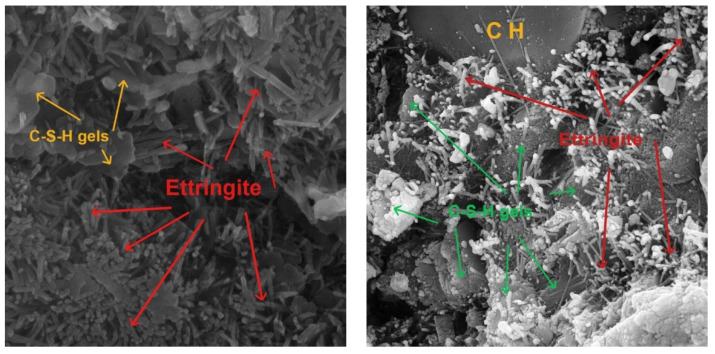
Cement hydration production.

**Figure 20 materials-16-05912-f020:**
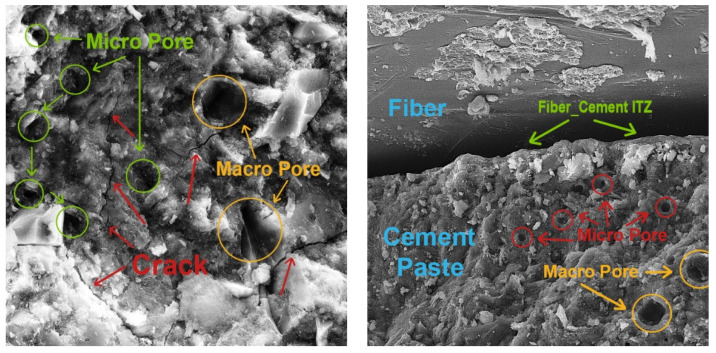
The cracks and pores in cement paste.

**Table 1 materials-16-05912-t001:** Binder chemical and physical composition [26].

Composition (%)	Cement	Metakaolin	Zeolite
CaO	≈63	≈1	≈8
SiO_2_	≈21	≈81	≈58
Al_2_O_3_	≈5	≈11	≈8
Fe_2_O_3_	≈4	≈1	≈2
MgO	≈2	≈1	≈4
K_2_O	0	≈1	≈1
Na_2_O	0	≈1	≈1
TiO_2_	0	≈0	≈0
MnO	0	≈0	≈0
L.O.I	≈4	≈3	≈19
Specific gravity (ASTM C188) [27]	3.15	2.45	2.18
Specific surface area (m^2^/kg) (Blain: ASTM C204) [28]	310	1200	320

**Table 2 materials-16-05912-t002:** Aggregate characteristics [26].

Aggregate	Water Absorption (%)	Specific Gravity
Fine	≈2.9	2.64
Coarse	≈1.8	2.42

**Table 3 materials-16-05912-t003:** Fiber characteristics [26].

Fiber Type	No	Width (cm)	Thickness (µm)	Diameter/ Length (cm)	Aspect Ratio	Density (kg/m^3^)	Tensile Strength (MPa)
Ring (R)	Ri1	0.5	30	≈6–6.5	-	≈680	≈570
	Ri2	1					
Strip (S)	St1	1	30	2.5	≈40		
	St2			5	≈80		

**Table 4 materials-16-05912-t004:** The concrete composition characteristics [26].

Mix	Sand (kg/m^3^)	Gravel (kg/m^3^)	Water (kg/m^3^)	Binders (kg/m^3^)			Fibers (kg/m^3^)					
				Cement (C)	Metakaolin (Mk)	Zeolite (Ze)	Ring (R)			Strip (S)		
							Ri1	Ri2		St1		St2
NF1	664	1054	205	369	41	-	-			-		
NF2	664	1054	205	328	82	-	-			-		
NF3	664	1054	205	369	-	41	-			-		
NF4	664	1054	205	328	-	82	-			-		
R1	664	1054	205	369	41	-	3.4		-	-		
R2	664	1054	205	369	41	-	-		3.4	-		
R3	664	1054	205	328	82	-	3.4		-	-		
R4	664	1054	205	328	82	-	-		3.4	-		
R5	664	1054	205	369	-	41	3.4		-	-		
R6	664	1054	205	369	-	41	-		3.4	-		
R7	664	1054	205	328	-	82	3.4		-	-		
R8	664	1054	205	328	-	82	-		3.4	-		
R9	664	1054	205	369	41	-	6.8		-	-		
R10	664	1054	205	369	41	-	-		6.8	-		
R11	664	1054	205	328	82	-	6.8		-	-		
R12	664	1054	205	328	82	-	-		6.8	-		
R13	664	1054	205	369	-	41	6.8		-	-		
R14	664	1054	205	369	-	41	-		6.8	-		
R15	664	1054	205	328	-	82	6.8		-	-		
R16	664	1054	205	328	-	82	-		6.8	-		
S1	664	1054	205	369	41	-	-			3.4	-	
S2	664	1054	205	369	41	-	-			-	3.4	
S3	664	1054	205	328	82	-	-			3.4	-	
S4	664	1054	205	328	82	-	-			-	3.4	
S5	664	1054	205	369	-	41	-			3.4	-	
S6	664	1054	205	369	-	41	-			-	3.4	
S7	664	1054	205	328	-	82	-			3.4	-	
S8	664	1054	205	328	-	82	-			-	3.4	
S9	664	1054	205	369	41	-	-			6.8	-	
S10	664	1054	205	369	41	-	-			-	6.8	
S11	664	1054	205	328	82	-	-			6.8	-	
S12	664	1054	205	328	82	-	-			-	6.8	
S13	664	1054	205	369	-	41	-			6.8	-	
S14	664	1054	205	369	-	41	-			-	6.8	
S15	664	1054	205	328	-	82	-			6.8	-	
S16	664	1054	205	328	-	82	-			-	6.8	

**Table 5 materials-16-05912-t005:** The chemical properties of water (Kg/m^3^) [26].

Chemical	PH	Hardness	Alkalinity	SO_4_ ^2−^	NO_2_^−^	NO_3_^−^	Cl^−^	Ca	Mg	NH_3_	Zn	Al	Cu	Fe
Lab water	≈6.7	45	20	100	0.01	2.82	5	4	≈0	≈0	≈0	0.3	0.02	≈0
Sea water	≈8	225	120	≈0	≈0	0.22	5	68	95	1.3	0.2	0.3	0.02	0.04

**Table 6 materials-16-05912-t006:** The weather conditions of Oman Sea tidal zone at experimental program time.

Chemical	March	April	May	June	July	August	September	October	November
Min temperature (°C)	27–21	31–27	31–29	30–27	30–27	28–27	28–24	24–20	21–17
Max temperature (°C)	31–26	35–30	37–32	33–30	34–31	33–31	33–29	31–28	30–24
Air humidity (%)	75–70	85–80	90–85	85–80	70–65	80–75	75–70	65–60	55–50

**Table 7 materials-16-05912-t007:** Maximum load capacity (P_max_).

Mix	P_max_ (kN)			Mix	P_max_ (kN)		
	28	90	180		28	90	180
N1	20.88	24.01	26.35	R15	13.08	15.07	17.68
N2	19.42	22.35	24.17	R16	13.73	16.27	19.67
N3	19.65	22.27	25.60	S1	15.31	17.97	20.33
N4	18.65	20.78	23.59	S2	16.87	19.57	21.95
R1	16.54	19.68	22.75	S3	13.92	16.17	18.09
R2	17.49	21.39	24.88	S4	15.66	18.12	20.54
R3	15.34	17.89	20.08	S5	14.79	17.52	20.98
R4	16.71	19.89	22.86	S6	15.21	18.46	22.66
R5	15.75	19.03	23.45	S7	13.23	15.30	18.30
R6	16.43	20.21	25.18	S8	14.45	17.30	21.03
R7	14.55	17.01	20.59	S9	14.05	15.98	17.91
R8	15.14	18.49	23.32	S10	14.67	16.92	18.86
R9	15.34	17.63	19.81	S11	12.20	13.94	15.35
R10	15.98	18.60	21.03	S12	13.71	15.60	17.32
R11	13.73	15.32	16.78	S13	12.76	15.23	18.44
R12	14.79	17.13	19.06	S14	13.17	15.83	19.59
R13	14.19	16.93	20.29	S15	11.91	13.63	16.06
R14	14.76	17.07	22.46	S16	12.86	14.92	17.83

**Table 8 materials-16-05912-t008:** Flexural toughness (T).

Mix	T (kN.m)			Mix	T (kN.m)		
	28	90	180		28	90	180
N1	2.40	2.64	2.84	R15	23.97	28.21	33.90
N2	2.13	2.30	2.46	R16	26.88	32.14	39.37
N3	2.28	2.46	2.67	S1	25.92	29.17	32.33
N4	2.06	2.17	2.35	S2	29.47	33.37	37.40
R1	22.24	24.93	27.20	S3	23.28	25.76	28.00
R2	24.77	27.73	30.58	S4	27.44	30.71	33.93
R3	20.40	23.03	25.11	S5	23.79	27.96	33.49
R4	23.71	26.58	28.69	S6	27.16	32.51	39.50
R5	20.77	23.91	28.09	S7	21.87	24.89	28.79
R6	23.22	27.42	33.05	S8	24.76	29.43	35.41
R7	19.35	22.02	25.39	S9	34.06	39.39	44.72
R8	21.58	24.70	29.02	S10	39.86	46.65	53.77
R9	28.70	32.85	36.80	S11	30.35	34.73	39.09
R10	33.08	38.16	43.55	S12	35.33	40.78	46.25
R11	24.97	28.34	31.60	S13	30.66	37.26	45.66
R12	28.46	33.33	37.75	S14	35.36	43.70	55.44
R13	25.94	30.91	37.18	S15	27.43	32.63	39.75
R14	29.55	36.14	44.94	S16	31.34	38.50	47.50

## Data Availability

Not applicable.

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
