# Peer review of "Investigating the Effects of Recycled Plastic as Fibers on Bending Behavior of Green Concrete Beams Exposed to Marine Environment"

_materials, 2023, doi:10.3390/ma16175912_

Round 1
Reviewer 1 Report
The present manuscript provided a comprehensive evaluation, highlighting the need for more detailed explanations in certain sections. The manuscript should incorporate additional information to address the concerns raised. Specifically discussion of methodology, providing a more comprehensive description of the experimental procedures and the rationale behind our approach. Additionally, additional data and analysis to support the findings, addressing the reviewer's request for a more robust interpretation of the results.
1. There are already so many review has already been done in the past.....What is the exact novelty of this current study... I suggest including a novelty statement in the Abstract section. Also, reduce the size of the abstract section.
2. Less literature is available related to plastic fiber. Authors are suggested to include quality literature.
3. Remove this extra full stop at Page no. 2 and line no. 66. Also, use citation number after the author's name Like... Yap et al. [15], Use 15 number just after the author's surname.
4. Authors should include the following recently published research papers in the introduction part, which are related to the sustainable use of waste.
-https://doi.org/10.1016/j.conbuildmat.2021.125730
-https://doi.org/10.1016/j.matpr.2020.10.829
5. Authors are suggested to mention respective codes for all physical properties of all materials.
6. Authors should provide the more details about the raw materials used in the study.
7. Can authors explain how ettringite is formed and, I also want to see more details about the SEM test.
8. Authors should add a brief reason for this strength enhancement at the end of the discussion part.
1. Improve the grammatical error and sentence formatting throughout the whole manuscript.
Author Response
- There are already so many review has already been done in the past.....What is the exact novelty of this current study... I suggest including a novelty statement in the Abstract section. Also, reduce the size of the abstract section.
- The exact novelty of this manuscript is the operation of recycled fibers and SCMs in concrete mixtures for using green concrete in marine structures, improving the mechanical properties of concrete structures, and reducing amount of air, marine, and environmental pollution. All of the novelties and aims are mentioned in the abstract, but the abstract is edited and the size of the abstract section is reduced.
- Less literature is available related to plastic fiber. Authors are suggested to include quality literature.
- Three manuscripts with recycled plastic fiber topics are added to the introduction section and the literature is edited.
- Remove this extra full stop at Page no. 2 and line no. 66. Also, use citation number after the author's name Like: Yap et al. [15], Use 15 number just after the author's surname.
- Your suggestion is done and reference numbers are edited.
- Authors should include the following recently published research papers in the introduction part, which are related to the sustainable use of waste.
A- doi.org/10.1016/j.conbuildmat.2021.125730
B- doi.org/10.1016/j.matpr.2020.10.829
- Your suggested manuscripts are added to the introduction.
- Authors are suggested to mention respective codes for all physical properties of all materials.
- Your suggestion is done.
- Authors should provide the more details about the raw materials used in the study.
- The details of raw materials (cement, SCMs, water, aggregates, fibers, etc.) for preparing concrete mixtures are presented in the experimental program section.
- Can authors explain how ettringite is formed and, I also want to see more details about the SEM test.
- Ordinary Portland cement is manufactured by a process that combines sources of lime (such as limestone), silica and alumina (such as clay), and iron oxide (such as iron ore). After grinding and heating the components, cement clinker is created. After cooling, the clinker is combined and ground with about 5% of calcium sulfate like gypsum (to regulate early hydration reactions, prevent flash setting, control setting time, strength development, and drying shrinkage potential) to form Portland cement. Gypsum and other sulfate compounds react with calcium aluminate in the cement to form ettringite (white, needle-like crystals) within the first few hours after mixing with water. Ettringite is the mineral name for calcium sulfoaluminate (3CaO•Al2O33CaSO4•32H2O). Most of the sulfate in the cement is normally consumed to form ettringite at early ages. The formation of ettringite in fresh concrete is the mechanism that controls stiffening. At this stage, ettringite is uniformly and discretely dispersed throughout the cement paste at a submicroscopic level. It is a necessary and beneficial component of Portland cement systems. After a few days, depending on the ratio of aluminum oxide to Portland cement sulfate, ettringite becomes unstable and turns into hydrated mono-sulfate (hexahedral sheets).
- More details for SEM tests are exhibited.
- Authors should add a brief reason for this strength enhancement at the end of the discussion part.
- Your suggestion is done and a brief reason for strength enhancement at the end of the discussion part is presented.
- Improve the grammatical error and sentence formatting throughout the whole manuscript.
- I tried to edit and improve the grammatical error and sentence formatting of the manuscript.

Reviewer 2 Report
The manuscript of the article "Investigating the Effects of Recycled Plastic as Fibers on Bending Behavior of Green Concrete Beams Exposed to Marine Environment" is devoted to the development of concrete with improved characteristics using recycled plastic (disposable cups). The novelty of the research lies in the limited number of scientific studies related to the operation of such "green" concrete in seawater.
The research has practical significance and can be recommended for publication after making significant adjustments:
1. The introduction should more clearly state the aims, objectives and scope of the study. Perhaps the chosen amounts of zeolite and metakaolin and the chosen ratios of materials in the concrete should have been justified in more detail.
2. In Table 1, replace words with formulas. It is desirable to merge Table 1 and 2. Information on the phase composition of zeolite and metakaolin is missing. Methods and equipment used for analysis are missing. How and by what methods were density and specific surface determined?
3. Figure 1 should be moved to the section "Results and their discussion".
4. Although certain regularities of concrete properties from fiber geometry (ribbon, ring) have been identified, the paper lacks a detailed characterization of the materials used for plastic processing. Providing information on the types of polymers, their sources could improve the reproducibility of the results.
5. Figure 4 should be removed from the paper altogether, as it is general in nature and provides absolutely no new scientific information.
6. The curing and curing conditions used to model the tidal zone conditions are not described in sufficient detail. Additional information would allow other researchers to more accurately replicate the test conditions.
7. In the reviewer's opinion, the article contains an excessive number of figures that complicate the interpretation of the data. The identification of key patterns should be dealt with separately. It is possible to reduce the number of figures and emphasize the most significant properties of the material.
8. SEM images are presented in the paper, but they are mentioned very superficially in the text of the paper, there is practically no analysis, which does not correlate well with the specifics of the journal.
9. The conclusions also need to be revised, as they state more facts. It is necessary to shift the emphasis on the relationship between changes in the functional properties of concrete and applied recommendations.
Author Response
- The introduction should more clearly state the aims, objectives and scope of the study. Perhaps the chosen amounts of zeolite and metakaolin and the chosen ratios of materials in the concrete should have been justified in more detail.
- Some manuscripts are added to the introduction and the introduction is edited. Furthermore, the used references for the chosen ratios of materials are presented.
- In Table 1, replace words with formulas. It is desirable to merge Table 1 and 2. Information on the phase composition of zeolite and metakaolin is missing. Methods and equipment used for analysis are missing. How and by what methods were density and specific surface determined?
- Tables 1 and 2 are merged and the formulas of composition are edited in the final Table.
- - Specific gravity by ASTM C188
- - Specific surface area by ASTM C204 (Blain)
- Figure 1 should be moved to the section "Results and their discussion".
- Figure 1 was removed from the manuscript.
- Although certain regularities of concrete properties from fiber geometry (ribbon, ring) have been identified, the paper lacks a detailed characterization of the materials used for plastic processing. Providing information on the types of polymers, their sources could improve the reproducibility of the results.
- Your suggestion is done and the details of the fibers are presented in the experimental program.
- Figure 4 should be removed from the paper altogether, as it is general in nature and provides absolutely no new scientific information.
- Figure 4 was removed from the manuscript.
- The curing and curing conditions used to model the tidal zone conditions are not described in sufficient detail. Additional information would allow other researchers to more accurately replicate the test conditions.
- Based on your suggestion, Figure 2 and Table 6 were added to explain the details of concrete curing conditions and the weather conditions of the sea tidal zone.
- In the reviewer's opinion, the article contains an excessive number of figures that complicate the interpretation of the data. The identification of key patterns should be dealt with separately. It is possible to reduce the number of figures and emphasize the most significant properties of the material.
- Your suggestion is done.
- SEM images are presented in the paper, but they are mentioned very superficially in the text of the paper, there is practically no analysis, which does not correlate well with the specifics of the journal.
- Analysis of SEM images is presented in section 5.2.
- The conclusions also need to be revised, as they state more facts. It is necessary to shift the emphasis on the relationship between changes in the functional properties of concrete and applied recommendations.
- The conclusion of the manuscript is edited.

Round 2
Reviewer 1 Report
The authors have successfully addressed all the comments satisfactorily, So my final decision is the ACCEPT.
Author Response
- The authors have successfully addressed all the comments satisfactorily, So my final decision is the ACCEPT.
- Thank you very much for your decision and consideration.

Reviewer 2 Report
Dear Authors, thank you for the comments taken into account! As for the chemical composition and physical properties of starting materials, my comment was not about standards, but about specific equipment, brands of instruments used for analysis. In general, most of the comments have been taken into account.
Author Response
- Dear Authors, thank you for the comments taken into account! As for the chemical composition and physical properties of starting materials, my comment was not about standards, but about specific equipment, and brands of instruments used for analysis.
- Laboratory Oven, Compressive strength, and bending testing machine are presented.
- In general, most of the comments have been taken into account.
- Thank you very much for your decision and consideration.
